ecology/behaviour

unmanned aircraft systems, drones, counting variability, behavioural reaction, *Lama guanicoe*, terrestrial mammals

**Author for correspondence:**
Natalia M. Schroeder
e-mail: natalias@mendoza-conicet.gob.ar

# An experimental approach to evaluate the potential of drones in terrestrial mammal research: a gregarious ungulate as a study model

Natalia M. Schroeder[1,2], Antonella Panebianco[2], Romina Gonzalez Musso[3] and Pablo Carmanchahi[2]

[1]Instituto Argentino de Investigaciones de las Zonas Áridas, CONICET, CC 507, CP 5500 Mendoza, Argentina
[2]Grupo de Investigación en Eco-Fisiología de Fauna Silvestre (INIBIOMA-CONICET-AUSMA-UNCo), Pasaje de la paz 235, CP 8370 San Martín de los Andes, Neuquén, Argentina
[3]Asentamiento Universitario San Martín de los Andes, Universidad Nacional del Comahue, Pasaje de la paz 235, CP 8370, San Martín de los Andes, Neuquén, Argentina

 NMS, 0000-0001-6530-2227; AP, 0000-0001-7900-1118

Research on the use of unmanned aircraft systems (UAS) in wildlife has made remarkable progress recently. Few studies to date have experimentally evaluated the effect of UAS on animals and have usually focused primarily on aquatic fauna. In terrestrial open arid ecosystems, with relatively good visibility to detect animals but little environmental noise, there should be a trade-off between flying the UAS at high height above ground level (AGL) to limit the disturbance of animals and flying low enough to maintain count precision. In addition, body size or social aggregation of species can also affect the ability to detect animals from the air and their response to the UAS approach. To address this gap, we used a gregarious ungulate, the guanaco (*Lama guanicoe*), as a study model. Based on three types of experimental flights, we demonstrated that (i) the likelihood of miscounting guanacos in images increases with UAS height, but only for offspring and (ii) higher height AGL and lower UAS speed reduce disturbance, except for large groups, which always reacted. Our results call into question mostly indirect and observational previous evidence that terrestrial mammals are more tolerant to UAS than other species and highlight the need for experimental and species-specific studies before using UAS methods.

# 1. Introduction

The use of unmanned aircraft systems (UAS, also called drones) in wildlife monitoring has grown dramatically in recent years [1]. The UAS are tools with clear comparative advantages over classic methods used in wildlife research, such as ground-based counts and manned aircraft. UAS are safe for operators, relatively less noisy than manned aircraft and able to access dangerous or remote areas. They allow repeating the same flight plan over time, capturing images with user-defined resolutions and carrying more than one sensor (e.g. a thermal camera) [2], considerably improving the quality of the data obtained [3]. Moreover, images provide a permanent recording of data that can be re-visited and analysed again in the future for unforeseen research questions. However, there are some limitations that still need to be overcome, such as a delay of legislation regarding technological progress in UAS and their rapid adoption as a research tool [4–7].

Research on the use of drones in wildlife is still in the trial phase, although there has been a remarkable progress in recent years. This advance has been very important in terms of knowing the technical feasibility of using different types of UAS (fixed-wing or multi-rotor) to detect, identify and count species and individuals with both optical and thermal sensors [3,8–12]. However, few studies have experimentally evaluated the effect of UAS on animals, while those studies done to date have focused primarily on aquatic fauna [13–17]. Both marine mammals [13] and aquatic birds [14,15] seem to tolerate the proximity of UAS flights so close that they allow collection of detailed morphometric data [18]. Observational studies have suggested that terrestrial mammals may respond less to UAS than aquatic fauna [19], but recent experimental evidence calls into question this claim. Ditmer *et al*. [20] observed strong physiological, but not behavioural, responses of bears to UAS flights, and Bennitt *et al*. [21] reported a wide range of reaction responses in herbivores, depending on the target species and UAS proximity. By analysing YouTube videos uploaded by laypeople, Rebolo-Ifran *et al*. [22] found that wildlife which uses terrestrial or aerial habitats are more likely to show a behavioural response to UAS than those occupying aquatic habitats, although the authors acknowledge that scientific information is still not conclusive. The increasing use of drones in both research and recreation calls for more studies that specifically evaluate the impact of UAS on a wide range of species and environments.

Each environment and species of interest represents a particular challenge for the use of drones that have guided research programmes. For example, in forested environments where animals are often difficult to detect visually, or to study species of nocturnal habits, it has been important to prioritize studies that evaluate the use of thermal sensors mounted on UAS to detect animals by their body heat [23–25]. For the study of marine species that live in large colonies, the priority has been to adjust automatic counting methodologies that minimize post-processing costs [8–10]. In arid open environments with relatively good visibility to detect animals but little environmental noise, there may be a trade-off between flying the UAS high in order to diminish the disturbance owing to noise level of the drone motors, while maintaining count precision in the images. In turn, it is expected that factors such as body size will affect the detectability of individuals from the air, and that group size will influence the warning and flight reactions to UAS approach of social species that benefit from grouping as a collective vigilance strategy [26,27]. To fill this information gap, we used an experimental approach applied to a native gregarious ungulate, the guanaco (*Lama guanicoe*) as a study model. We aimed to (i) evaluate the variation in counts of adult and offspring in images taken at different UAS heights above ground level (AGL) and (ii) assess the behavioural reaction of guanacos before and during approaching UAS, and at different combinations of UAS height AGL, speed and animal group size.

# 2. Methods

This work was carried out in two study areas of the centre-west of Argentina: (i) La Payunia Provincial Reserve (−36°36′ S, −68°34′ W) in Mendoza province, supporting about 26 000 wild guanacos [28] and (ii) Los Peucos Farm (−39°43′ S, −71°03′ W) in Neuquén province, housing 400 guanacos in captivity, where animals can move freely in enclosures of 6 km². We used the Phantom 4 Advance (DJI, Shenzhen, China), a small quadcopter with an onboard 20-megapixel camera. The technical specifications can be found at https://www.dji.com/phantom-4-adv/info#specs.

## 2.1. Flight plans and data record

To evaluate our objectives, we combined three types of flights (table 1 and figure 1) carried out by two people, i.e. a drone pilot and another person who recorded animal behaviour when necessary.

**Table 1.** Details of UAS flight types performed by location and objective.

| date | flight type | location | objective | number of flights | flight height AGL (m) | number of pictures gathered | number of pictures used for counting |
|---|---|---|---|---|---|---|---|
| Nov 2017 | vertical | Payunia | count | 1 | 150 | 1 | 1 |
| | | | | | 200 | 1 | 1 |
| Sep 2018 | | Peucos | | 2 | 50 | 2 | 2 |
| | | | | | 100 | 2 | 2 |
| | | | | | 150 | 2 | 2 |
| | | | | | 200 | 2 | 2 |
| Feb 2018 | horizontal | Payunia | count, behaviour | 9 | 60 | 11 | 11 |
| | | | | 17 | 180 | 22 | 22 |
| | | | behaviour | 38 | 60 | 0 | 0 |
| | | | | 27 | 180 | 0 | 0 |
| Nov 2017 | scanning | Payunia | count | 1 | 200 | 24 | 9 |
| | | | | 3 | 150 | 58 | 27 |
| Sep 2018 | | Peucos | | 1 | 100 | 33 | 28 |
| Feb 2018 | scanning behaviour | Payunia | count, behaviour | 3 | 200 | 36 | 18 |

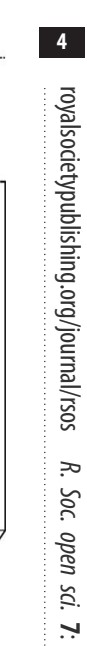

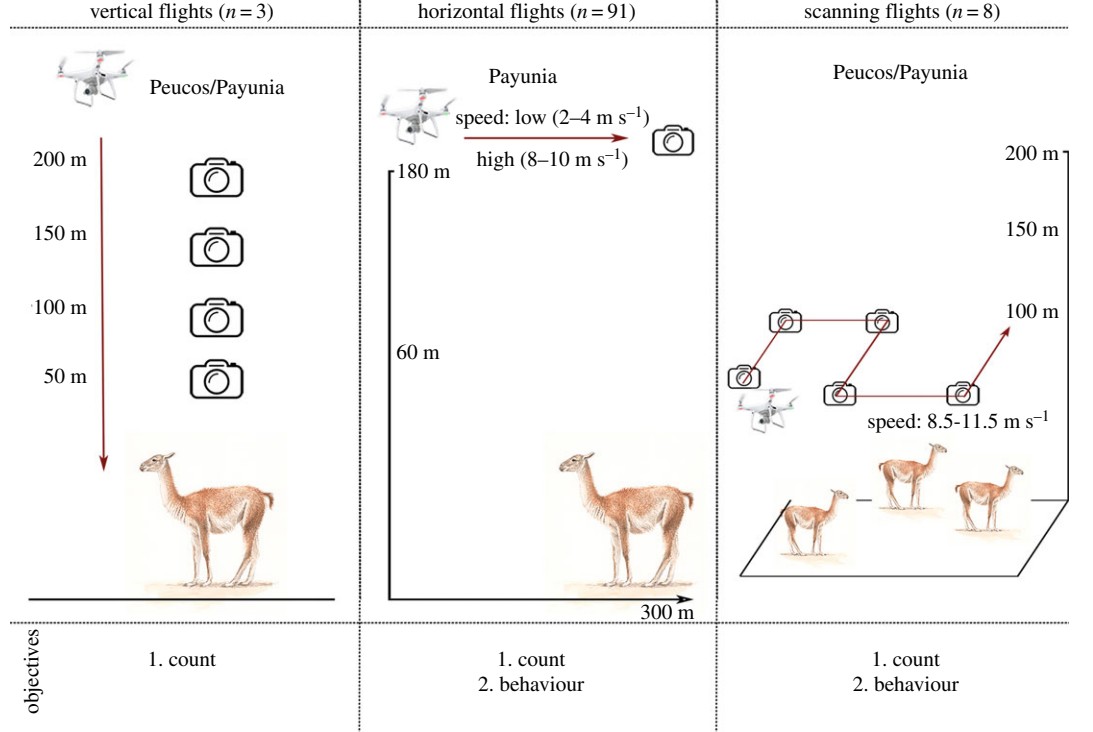

**Figure 1.** Flight plans performed in Los Peucos Farm and La Payunia Reserve. The camera symbol indicates a photograph being taken.

### 2.1.1. Vertical flights

We approached the drone at 200 AGL until it was positioned directly above a previously selected group of guanacos. Then the drone was slowly lowered and pictures were taken at 200, 150, 100 and 50 m AGL.

### 2.1.2. Horizontal flights

We performed horizontal flights only in the wild population (figure 1 and table 1). We chose a guanaco group at least 300 m away from the observer and flew the drone towards it at constant height and speed. We combined low (2–4 m s$^{-1}$) and high (8–10 m s$^{-1}$) speeds with low and high heights (60 and 180 m AGL, respectively). We observed all the animals as a focal group [29] and recorded whether there was a flight reaction of the animals before the drone was positioned above the group (figure 1). Behavioural observations were made from the ground using binoculars (10 × 42 mm; Vanguard), a 60 mm spotting scope (20–60×; Bushnell Trophy XLT) and registered using a digital recorder (Panasonic RR-US551). In order to prevent the habituation of guanacos to the approaching drone, we drove along existing roads at the Reserve to prevent repeat sampling of the same focal group on the same day and conducted each trial in different areas of La Payunia Reserve. When groups were disturbed by human presence, flights and observations were not initiated. The observer recorded group size in the field by counting the number of adults and offspring, based on body size. Groups were identified by excluding, at the beginning of each observation, individuals more than 300 m away from their neighbours. In each case, this was confirmed by the movement of the animals during the observations (i.e. the members of the same group moved together in the same direction, while the other individual(s) stayed in the same place or moved in another direction) [26,30]. We obtained pictures at 60 and 180 m AGL with the camera facing downwards, of those groups of animals that did not react to the approaching drone.

### 2.1.3. Scanning flights

We conducted pre-programmed autonomous flights (using PIX4D software) over a previously defined square area, following parallel line transects. We obtained pictures with the camera facing downwards. Flights were scheduled at 100 m AGL (10 m s$^{-1}$ of speed), 150 m (8.5 m s$^{-1}$) and 200 m

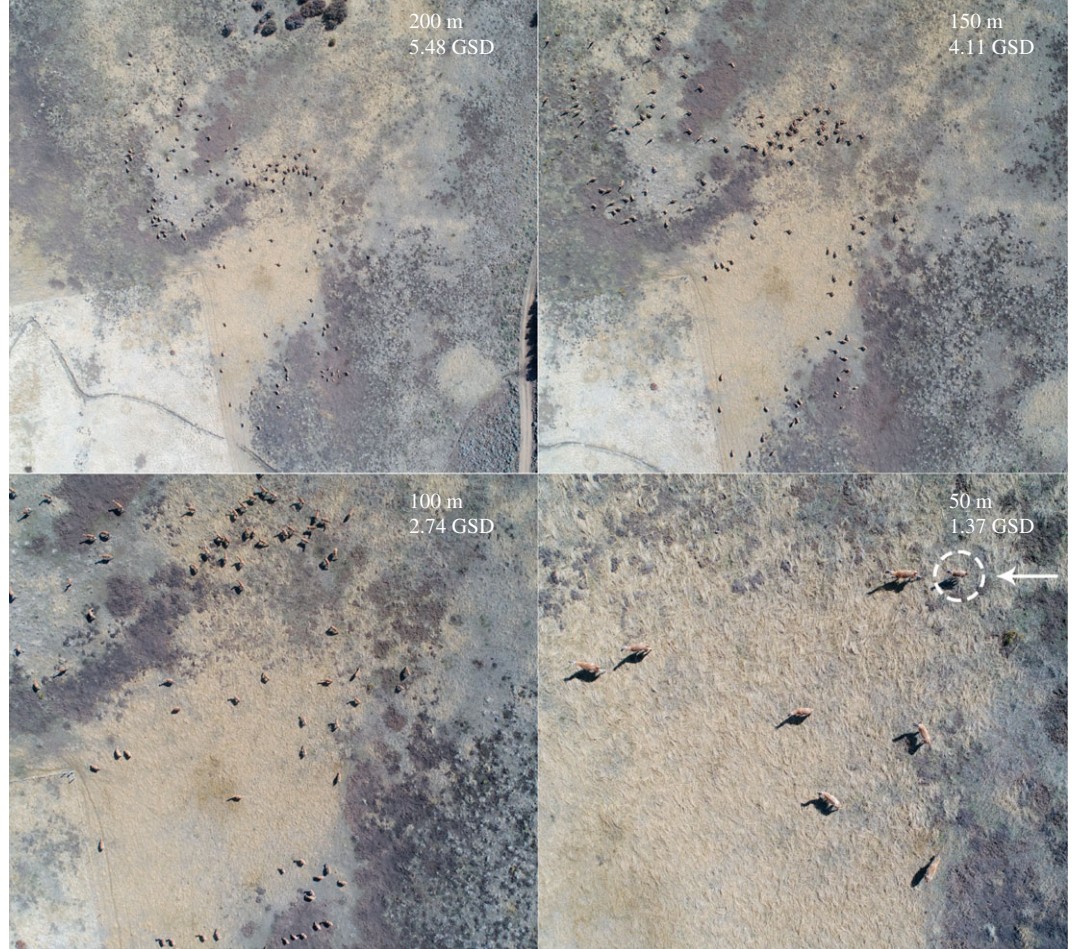

**Figure 2.** UAS imagery of a group of guanacos at 200, 150, 100 and 50 m AGL (GSD, ground sampling distance). The arrow indicates a calf behind an adult.

(8.5–11 m s$^{-1}$), with minimum percentage of overlap between images (20% frontal/20% lateral) and covering areas between 20.25 and 44.8 ha. In three of these scanning flights (named scanning behaviour flights, table 1), we flew at 200 m AGL above previously identified groups of guanacos to record the behaviour (foraging, vigilance, locomotion, others) of each of the members of the group before and during the UAS mission and following the same flight path of the drone. To do this, we used a scan sampling methodology [29]. Behavioural observations and group size were registered as described in horizontal flights. Scanning flights were performed in both wild and captive populations, and scanning behaviour flights only in the wild population (figure 1 and table 1).

## 2.2. Data analysis

### 2.2.1. Counting variability

We selected photos from all flight plans at 50–60, 100, 150, 180, and 200 m AGL, totalling 125 images (table 1). Four independent and trained observers manually checked the images following the same protocol, which consisted of overlaying a grid on each image, zooming in if necessary, observing the image from left to right and from top to bottom looking for guanacos, marking them and recording the total number of individuals and the number of offspring. The young are distinguishable by their smaller size and their proximity to their mother (figure 2). All the images were analysed in the software GIMP 2.10.6 (https://www.gimp.org/). The images were randomly arranged within each height class so that observers would not review images of the same flight plan consecutively. Coefficients of variation (CV = standard deviation/mean) of the counts per photo were obtained. The Kruskal–Wallis (K-W) $\chi^2$ and pairwise Wilcoxon tests were used to compare medians of the CV at different heights, expecting higher counting variance at higher heights.

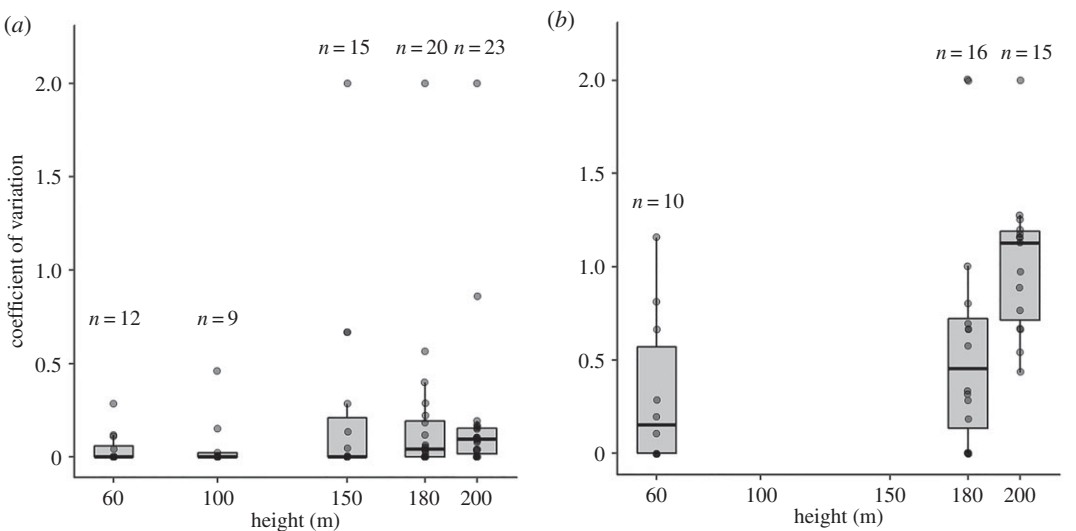

**Figure 3.** Coefficients of variation (CV) at different heights of the (*a*) total counts and (*b*) offspring counts. The numbers above the bars indicate the number of photos with counts, by height. The dots indicate the CV of each photo.

### 2.2.2. Behavioural reaction

We classified guanaco responses from horizontal flights as (i) flight reaction (walking quickly or running away from the guanaco's original location in the opposite direction to the UAS), or (ii) no reaction, which included apparent lack of detection (the animals continued displaying the same behaviour recorded before the UAS flight) or UAS detection (alert posture, with the animal standing with its head and neck upright, ears erect and aimed directly at the stimulus) [30]. We considered a flight reaction when at least one of the individuals of the focal group behaved that way, followed by the others. From scanning behaviour flights, we analysed changes in the percentage of each behavioural category considered before and during the UAS mission calculated as the number of animals displaying each behavioural category divided by the total number of animals within each group.

## 3. Results

### 3.1. Counting variability

Observers could count adults and offspring in photos taken at all heights (figure 2). No differences were found between the median CV of the total counts between different observers counting from the same images at different heights (K–W $\chi_4^2 = 6.02$, $p = 0.2$), although a greater range of variation of CVs per photo and extreme values were observed at higher heights (figure 3*a*). By contrast, the median CV of offspring counts differed between heights (K–W $\chi_2^2 = 12.6$, $p = 0.002$), with CV of counts at 200 m larger compared to 50/60 m (Wilcoxon rank test $p = 0.004$) and 180 m (Wilcoxon rank test $p = 0.025$) (figure 3*b*). At 100 and 150 m AGL, only a small number of photos were captured that contained offspring, so we could not include these images in the analysis.

### 3.2. Behavioural reaction in horizontal flights

Out of 47 experiments at 60 m AGL, 87% of the groups ($n = 41$) reacted before the drone was positioned above them. Of these, 51.2% of the groups reacted at 2–4 m s$^{-1}$, while 48.8% reacted at 8–10 m s$^{-1}$. At 180 m AGL, out of 44 experiments, about half of the groups (47.7%, $n = 21$) reacted before the UAS reached them, of which 42.9% did so at 2–4 m s$^{-1}$, while 57.1% did so at 8–10 m s$^{-1}$ of speed (figure 4*a*). Globally, at lower heights, the speed did not influence the reaction of the animals, but at higher heights, 14% more groups reacted when the UAS was flown at 8–10 m s$^{-1}$. The groups that did not react to the advance of the drone ranged from 1 guanaco (60 m, both forward speeds), 1.4 (180 m/2–4 m s$^{-1}$), and 2.44 guanacos on average (180 m/8–10 m s$^{-1}$). On the other hand, the groups that reacted before the drone was positioned above them had 17.5 (60 m/2–4 m s$^{-1}$), 15.9 (60 m/8–10 m s$^{-1}$), 38.7 (180 m/2–4 m s$^{-1}$) and 17.3 (180 m/8–10 m s$^{-1}$) guanacos on average. All groups of

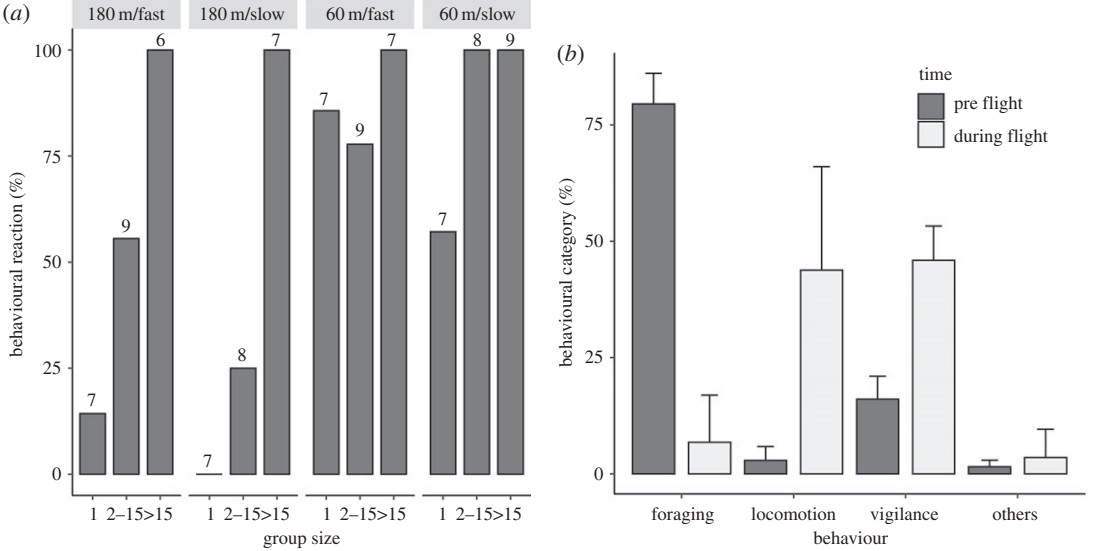

**Figure 4.** Behavioural reaction of guanacos. (*a*) Percentage of reaction of three categories of guanaco group size at different combinations of height and speed (slow: 2–4; fast: 8–10 m s$^{-1}$) during horizontal flights. Numbers above the bars represent the sample size in each combination of height/speed. (*b*) Average (and s.d.) of the percentages of behavioural categories recorded before and during the scanning behaviour flights.

greater than 15 animals reacted to the UAS ($n = 38$ animals per group on average), irrespective of flight height and speed. The combination of 180 m/2–4 m s$^{-1}$ generated the least reaction for groups of up to 15 guanacos (figure 4*a*).

## 3.3. Behavioural reaction in scanning behaviour flights

Group sizes in scanning behaviour flights were 17, 38 and 50 animals. On average, before UAS missions, the most common behavioural category was foraging (79.52%), followed by vigilance (16.06%) and locomotion (2.87%). During UAS flights, most individuals changed foraging behaviour to vigilance and locomotion (45.90% and 43.79%, respectively; figure 4*b*).

# 4. Discussion

Ongoing monitoring of terrestrial animal populations, especially of large-bodied species using extensive areas, has been a constant logistical challenge for researchers and wildlife managers. The emergence of UAS as an innovative tool for aerial monitoring of terrestrial fauna presents itself as promising, but it still requires more studies testing its potential use, applications and impact across different species and landscapes. To our knowledge, our study is the first published paper on this topic for South America [22]. Here, we provide key information related to animal counting variability from imagery and UAS impact on a gregarious species that can be applied to improve this novel technology as a tool for monitoring terrestrial mammals.

The variation in counting animals increased at higher flight heights only for guanaco offspring, as was expected owing to their smaller size compared to adults. The high variability in offspring counts indicates that observers made more counting errors, i.e. false positives (detecting offspring where there are none) and false negatives (not detecting offspring where there are). Contrasting effects of surrounding vegetation, time of the day or animal postures (e.g. an adult feeding with the neck down may resemble an offspring in size) surely contributed to the observers' misidentification. Further training of observers along with the use of digital tools could aid not only in decreasing possible errors in counting offspring but also in overcoming the limitations of the considerable effort required to process images. In this regard, there are recent promising examples of the use of special software to assist with manually counting animals in images [31], such as semi-automated identification systems used to count gulls (*Larus fuscus*) [15], and automatic identification and counting algorithms applied to hippos (*Hippopotamus amphibius L*) [32] and koalas (*Phascolarctos cinereus*) [33]. Furthermore, although it is a developing field, the counting of individuals based on UAS imagery has been shown to improve accuracy compared to ground-based counts [3,15,34].

Guanacos reacted more to UAS at lower heights at any combination of speed, but at higher heights, faster drones elicited more reactions of small-to-medium-sized groups. The guanaco has excellent peripheral vision and a wide field of view that gives it sharp distant vision [35]. This, combined with elongated and highly mobile ears, allows the guanaco to locate movement at a certain distance to detect its main predator, *Puma concolor*, which has a stalking hunting strategy. In our study, the animals reacted at heights high enough to make visual detection of the UAS unlikely, so it is possible that the behavioural responses were triggered by auditory rather than visual signals, which supports previous research suggesting other large herbivores also react more to auditory disturbance [21]. This is supported by the fact that the guanaco reaction was greater at higher UAS speeds, when the noise level of the motors is greater. In turn, all groups of more than 15 individuals reacted, regardless of the combination of height and speed. During the UAS scanning behaviour flights, even with a low sample size, all large groups showed a clear change in behaviour, from foraging to vigilance and locomotion. These results may be related to the cooperative vigilance strategy described for this social species [26] and the increased probability of detecting and reacting to a threat in large groups [29]. However, personal observations suggest that the behavioural response of guanaco to UAS is not similar in captive animals, which appear to be less reactive. It is expected that this may also occur in those wild populations usually exposed to human presence (e.g. tourism) acting as a neutral stimulus for enough time [36–38]. It would be necessary to replicate these experiments in other populations that have experienced different histories of anthropogenic disturbance to elucidate broader patterns of guanaco–UAS relationships across the species' range. Additionally, further research is needed to evaluate other possible negative impacts that do not have a behavioural correlate, such as physiological responses, and should also consider how UAS response could be influenced by other variables, such as the number of offspring in the group [26], habitat structure and wind noise, and ultimately assess if repeated exposure to UAS flights could elicit habituation [39]. For the purpose of sampling design of guanaco populations, it seems that flying at high height (180–200 m AGL) and at low speed (2–4 m s$^{-1}$) during the reproductive season when animals are aggregated in small groups (less than 15–20) could be the best combination to achieve a balance between efficiently detecting guanaco adults (and probably offspring with the help of digital tools) and minimizing disturbances. This sampling protocol could be posed as a hypothesis to be tested on other gregarious terrestrial mammals living in open habitats.

Overall, our study species showed a high sensitivity to the UAS approach. Eighty-seven per cent of the groups analysed at 60 m AGL reacted before the drone approached them, and 48% reacted at 180 m AGL. By simulating recreational UAS approaches (vertical and horizontal flights at maximum speed), Bennitt *et al.* [21] concluded that the response strength to UAS depends strongly on the target species and UAS proximity. Our study provides new evidence on the sensitivity of terrestrial mammals to UAS, paying special attention to grouping as a factor affecting the reaction responses of social species. These results call into question previous mostly indirect and observational evidence that terrestrial mammals are more tolerant to disturbance by UAS than, for example, birds [19] and highlight the need for further experimental and species-specific studies to assess the potential impacts on the target species before starting to use UAS as a research tool and to make more robust generalizations. Our findings on guanacos in particular are also especially important because they alert management agencies and users in general about the potential effects in the near future of recreational activity based on drones, an activity still under-developed in South America compared to other continents [22] but that is growing worldwide.

Ethics. The study was carried out under the permits RESOL-2019-154-E by the Natural Resources Department from the Government of Mendoza, and ANAC Nro 00005366 and IF-2019-68620267-APN-DNSO#ANAC from the National Civil Aviation Administration.

Data accessibility. Data used for all analyses are available as the electronic supplementary material.

Authors' contributions. N.M.S. conceived the study and obtained funds, N.M.S. and A.P. collected field data, carried out the design of the study and statistical analyses, and drafted the manuscript; R.G.M. collected field data and critically revised the manuscript; P.C. helped with UAS funds, and critically revised the manuscript; N.M.S., A.P., P.C. and R.G.M. obtained permits. All authors gave final approval for publication and agree to be held accountable for the work performed therein.

Competing interests. We declare we have no competing interests.

Funding. This work was supported by grant no. PICT 2015 1780 from ANCyT-FONCYT and grant no. PIP 2015 0425 from CONICET and CIEFAP (Argentina).

Acknowledgements. We thank M. Palma, E. Soto, M. Vázquez, N. Manfre, N. Vivanco, P. Gregorio and B. Hoepke for their field support. We also thank P. Gregorio and A. Marozzi for her assistance in image counting, and Mary Rowland for her support in the English writing style.

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
