## [Reviewer comments · Royal Society Open Science]

Review History

RSOS-191482.R0 (Original submission)

Review form: Reviewer 1

Is the manuscript scientifically sound in its present form?

No

Are the interpretations and conclusions justified by the results?

Yes

Is the language acceptable?

No

Do you have any ethical concerns with this paper?

No

Have you any concerns about statistical analyses in this paper?

No

Recommendation?

Major revision is needed (please make suggestions in comments)

Comments to the Author(s)

This paper presents the results of several experiments to determine the reaction of guanaco to drones, and the variation of manual counts of these animals based on photos taken from the drone. Both of these tests were performed for a variety of drone heights and speeds of approach to the animals. The experimental design with regards to the drone operations themselves is ok, but the analysis needs more work to be of publishable quality. The methods and results section are somewhat unclear and lack necessary detail. It's true that there is not a lot of previous work in the reactions of animals to drones; and this reviewer agrees that it is likely species specific, and subject to many factors, and that we should be making note of animal reactions and adjusting observing strategies accordingly. Overall this paper does provide some new insight which may be of use to the community, provided some revisions are made.

The authors have missed a significant amount of literature on this subject. The statement in the abstract that progress so far has "mainly been derived from aquatic fauna studies" is perhaps not accurate given the wealth of other studies on terrestrial animals, such as those listed at the end of my review.

Throughout the paper the authors make incorrect use of the word "altitude". Altitude has a specific meaning, which is the height above sea level. To say you flew the drone at 60m altitude at the co-ordinates specified would likely mean you were flying underground. All instances of the word altitude should be replaced with "height" or "height AGL". I know this might not sound very scientific but it is the correct term.

Clarity of method description needs to be improved throughout. More detail is required in table 1, each flight type needs split into heights and number of photos gathered. If no photos were taken on a flight - ie it was for behaviour observation only, this should also be indicated. It is not clear if behaviour observing flights were performed before, after or at the same time as photo gathering flights. It may be that the animals became more habituated to the drone in later flights so this needs to be made clearer. The specific speeds need to be clarified for the scanning flights rather than just the range. What was the overlap in photos taken on the horizontal flights?

For the behaviour observations, how were the observations made and by how many people?

Its not clear if one of the aims of the study was to count the total number of guanacos, or if you were only interested in the number per photo. If the total number was a value of interest this should also be indicated along with details of how you account for animals moving between photos in scanning flights.

The authors mention "accuracy" in the discussion. Since no "true" ground counts were made, or at least weren't reported if they were made, it is hard to make strong statements about accuracy. The variance between different observer's counts is what is reported, and this should be made clear. Its true that variation will be related to accuracy if taking averages, but since the authors don't know if there may be a systematic offset between true and observed counts, strong statements about accuracy should be removed.

Were there any false positives identified? Given the variation in counts this seems likely. This and possible causes of false positives should be discussed.

Why was the response of the animals not monitored during VF flights? Some previous papers find that animals react strongly to a descending drone.

Was there a difference in behaviours between the wild and captive groups?

The first paragraph of the results section needs clarifying. It could read as though the count variation you are reporting is between images of the same group at different heights (I'm fairly sure that's not what you mean). Needs to be clear that the variation is between different observers counting from the same images.

Line 47 - Drones are a type of remote sensing platform, presumably you are referring to satellites here?

Line 54 - its not clear what you mean by "sampling coverage capacity" or "low flight autonomy". These terms need to be defined more clearly or replaced.

Lines 58-61 - I'm not convinced this is true as there are many papers on terrestrial animals as well.

Line 64 - a rainforest environment is very noisy, are you sure that marine environments are louder? Are there any measurements to support this statement?

Line 111 - "y" -> 'and'

Line 163 - Its not clear how 87% arises from 2 groups of about 50% each. Needs to be clearer where this number comes from.

Line 169 - Does this mean 14% more groups or 14% more animals?

Line 200 - similar to above, the variation has increased but you can't really make strong statements about the probability of miscounting in an absolute sense.

Line 220 - 'motors' not 'engines'

Figures - more examples from different heights would help illustrate why and how it is harder to count animals at higher height AGL.

References to add

Wich, S.A.; Koh, L.P. Conservation Drones. Mapping and Monitoring Biodiversity; Oxford University Press: Oxford, UK, 2018.

Longmore, S.N.; Collins, R.P.; Pfeifer, S.; Fox, S.E.; Mulero-Pázmány, M.; Bezombes, F.; Goodwin, A.; De Juan Ovelar, M.; Knapen, J.H.; Wich, S.A. Adapting astronomical source detection software to help detect animals in thermal images obtained by unmanned aerial systems. *Int. J. Remote Sens.* 2017, 38, 2623–2638.

Mulero-Pázmány, M.; Stolper, R.; van Essen, L.D.; Negro, J.J.; Sassen, T. Remotely piloted aircraft systems as a rhinoceros anti-poaching tool in Africa. *PLoS One* 2014, 9, e83873.

- Seymour, A.C.; Dale, J.; Hammill, M.; Halpin, P.N.; Johnston, D.W. Automated detection and enumeration of marine wildlife using unmanned aircraft systems (UAS) and thermal imagery. *Sci. Rep.* 2017, 7, 45127.
- Kays, R.; Sheppard, J.; Mclean, K.; Welch, C.; Paunescu, C.; Wang, V.; Kravit, G.; Crofoot, M. Hot monkey, cold reality: Surveying rainforest canopy mammals using drone-mounted thermal infrared sensors. *Int. J. Remote Sens.* 2019, 40, 407–419.
- Van Andel, A.C.; Wich, S.A.; Boesch, C.; Koh, L.P.; Robbins, M.M.; Kelly, J.; Kuehl, H.S. Locating chimpanzee nests and identifying fruiting trees with an unmanned aerial vehicle. *Am. J. Primatol.* 2015, 77, 1122–1134.
- Bonnin, N.; Van Andel, A.; Kerby, J.; Piel, A.; Pintea, L.; Wich, S.; Bonnin, N.; Van Andel, A.C.; Kerby, J.T.; Piel, A.K.; et al. Assessment of chimpanzee nest detectability in drone-acquired images. *Drones* 2018, 2, 17.
- Wich, S.; Dellatore, D.; Houghton, M.; Ardi, R.; Koh, L.P. A preliminary assessment of using conservation drones for Sumatran orang-utan (*Pongo abelii*) distribution and density. *J. Unmanned Veh. Syst.* 2016, 4, 45–52.
- Burke, C.; Rashman, M.; Wich, S.; Symons, A.; Theron, C.; Longmore, S. Optimising observing strategies for monitoring animals using drone-mounted thermal infrared cameras. *Int. J. Remote Sens.* 2019, 40, 439–467.
- Burke, C.; Rashman, M.F.; Longmore, S.N.; McAree, O.; Glover-Kapfer, P.; Ancrenaz, M.; Wich, S.A. Successful observation of orangutans in the wild with thermal-equipped drones. *J. Unmanned Veh. Syst.* 2019, 00: 1–23 (0000) [dx.doi.org/10.1139/juvs-2018-0035](https://doi.org/10.1139/juvs-2018-0035).
- Witczuk, J.; Pagacz, S.; Zmarz, A.; Cypel, M. Exploring the feasibility of unmanned aerial vehicles and thermal imaging for ungulate surveys in forests – Preliminary results. *Int. J. Remote Sens.* 2018, 39, 5504–5521.
- Denise Spaan, Claire Burke, Owen McAree, Filippo Aureli, Coral E. Rangel-Rivera, Anja Hutschenreiter, Steve N. Longmore, Paul R. McWhirter, and Serge A. Wich. Thermal infrared imaging from drones offers a major advance for spider monkey surveys. *Drones*, 3(2), 2019.
- Chrétien, L.-P., Théau, J., and Ménard, P. 2015. Wildlife multispecies remote sensing using visible and thermal infrared imagery acquired from an unmanned aerial vehicle (UAV). *Int. Arch. Photogramm. Remote Sens. Spatial Inf. Sci.* XL-1/W4: 241–248. doi: 10.5194/isprsarchives-XL-1-W4-241-2015.
- Duffy, J. P., A. M. Cunliffe, L. DeBell, C. Sandbrook, S. A. Wich, J. D. Shutler, I. H. Myers-Smith, M. R. Varela, and K. Anderson. 2017. "Location, Location, Location: Considerations When Using Lightweight Drones in Challenging Environments." *Remote Sensing in Ecology and Conservation*. doi:10.1002/rse2.58.
- Gooday, O. J., N. Key, S. Goldstien, and P. Zawar-Reza. 2018. "An Assessment of Thermal-Image Acquisition with an Unmanned Aerial Vehicle (UAV) for Direct Counts of Coastal Marine Mammals Ashore." *Journal of Unmanned Vehicle Systems* 6 (2): 100–108. doi:10.1139/juvs-2016-0029.
- Hodgson, J. C., S. M. Baylis, R. Mott, A. Herrod, and R. H. Clarke. 2016. "Precision Wildlife Monitoring Using Unmanned Aerial Vehicles." *Scientific Reports* 6: 22574 EP. doi:10.1038/srep22574.

Israel, I., and A. Reinhard. 2017. Detecting Nests of Lapwing Birds with the Aid of a Small Unmanned Aerial Vehicle with Thermal Camera. ICUAS 2017- International Conference on Unmanned Aircraft Systems (ICUAS), Miami, FL, June 13-16.

Israel, M. 2011. "A Uav-Based ROE Deer Fawn Detection System." ISPRS - International Archives of the Photogrammetry, Remote Sensing and Spatial Information Sciences 3822: 51-55.

Evangeline Corcoran, Simon Denman, Jon Hanger, Bree Wilson & Grant Hamilton, "Automated detection of koalas using low-level aerial surveillance and machine learning", Scientific Reports volume 9, Article number: 3208 (2019)

Review form: Reviewer 2

Is the manuscript scientifically sound in its present form?

Yes

Are the interpretations and conclusions justified by the results?

Yes

Is the language acceptable?

Yes

Do you have any ethical concerns with this paper?

No

Have you any concerns about statistical analyses in this paper?

No

Recommendation?

Accept with minor revision (please list in comments)

Comments to the Author(s)

This study expands on our understanding of how to best implement, and the disturbance caused by, the use of UAS to census and study terrestrial mammal species. Studies such as these that specifically measure and test the disturbance to animals through different flight patterns, noise levels, and altitudes are important given the burgeoning use of UAS in wildlife research. The study is well conducted and the manuscript well written. There are a few minor changes that could be made to clarify the English and I've listed these more specifically below, but overall the introduction could contextualize the proposed research in more detail compared to what has been done to date and provide more detail and clarification in the methods to improve the manuscript.

Introduction

As mentioned, most research today has been done on marine mammals and birds, however, I think more could be included here about previous research into disturbance of UAS on different species, particularly that terrestrial mammals. For example Bennitt et al. 2019, Terrestrial mammalian wildlife responses to Unmanned Aerial Systems approaches, who found varying degrees of response several terrestrial mammal species to UAS. Can you highlight the necessary advance brought by this study in contrast to previous work? Also, research such as this that are

mentioned in conclusion could be raised earlier in introduction describing that studies to date suggest terrestrial mammals may respond less (Mulero-Pázmány M, Jenni-Eiermann S, Strebel N, Sattler T, Negro JJ, Tablado Z (2017) Unmanned aircraft systems as a new source of disturbance for wildlife: A systematic review. PLoS ONE 12(6): e0178448).

L48 in comparison to what? i.e. UAS are relatively less noisy than manned aircraft but not remote sensing

L51 I'm not sure I understand what is meant here, can you clarify? Perhaps that recording images using UAS offers an advantage over traditional ground surveys because the images are permanent and can be re-visited in the future for other, unforeseen analyses?

L58-59 Needs clarification and fixing for english

L61 "is" instead of "are"

Methods

L82 could you describe a bit more the enclosure particularly with relevance to their available space to react to the drones

L94 Why only 2 pictures taken at Payunia? Or is this number of pictures at each elevation for VF flights?

L104 how was group size counted? Here, and below in describing S flights, can you clarify how group size and behaviour was recorded? i.e. were there observers on foot, were the guanacos habituated to humans on the ground, or was this data collected using the imagery? If the latter, can you provide more detail as it sounds from L105 and 106 that images were only captured if the animals did not react.

L114 can you also provide more detail on the behavioural data? i.e. data categories (forage, vigilance etc., is mentioned at the group level – was this scored based on the majority of the individuals in the group or quantified in some way i.e. number of individuals doing each behaviour on each behavioural scan?

L134-139 as above, I think this needs some clarification on the unit of analysis (i.e. one measure of behaviour per group, or individuals?)

L140 can you clarify if this is using imagery?

L150 "with VC counts at 200 m found to be more variable compared to"

Discussion

Is it possible to highlight the importance and contribution of this study in contrast to Bennett et al. 2019's recent analysis? Perhaps by paying more attention to inter-species differences and why reproducing these studies across a range of species and ecotypes is necessary. Given that responses may vary by species, could you also provide some discussion of the importance of these findings that may be specific to Guanacos or to disturbance potentially caused by the increasing recreational use of drones (if this is also occurring across S. America)?

L202 perhaps "aid" would make more sense here than "collaborate"?

Decision letter (RSOS-191482.R0)

28-Sep-2019

Dear Dr Schroeder,

The editors assigned to your paper ("An experimental approach to evaluate the potential of drones in terrestrial mammal research: a gregarious ungulate as a study model") have now received comments from reviewers. We would like you to revise your paper in accordance with the referee and Associate Editor suggestions which can be found below (not including

confidential reports to the Editor). Please note this decision does not guarantee eventual acceptance.

Please submit a copy of your revised paper before 21-Oct-2019. Please note that the revision deadline will expire at 00.00am on this date. If we do not hear from you within this time then it will be assumed that the paper has been withdrawn. In exceptional circumstances, extensions may be possible if agreed with the Editorial Office in advance. We do not allow multiple rounds of revision so we urge you to make every effort to fully address all of the comments at this stage. If deemed necessary by the Editors, your manuscript will be sent back to one or more of the original reviewers for assessment. If the original reviewers are not available, we may invite new reviewers.

- Data accessibility

If you wish to submit your supporting data or code to Dryad (<http://datadryad.org/>), or modify your current submission to dryad, please use the following link:
<http://datadryad.org/submit?journalID=RSOS&manu=RSOS-191482>

- Competing interests

- Authors' contributions

- Acknowledgements

- Funding statement

on behalf of Dr Alecia Carter (Associate Editor) and Kevin Padian (Subject Editor)
openscience@royalsociety.org

Associate Editor's comments (Dr Alecia Carter):

Dear authors,

I have now received two reviews of your manuscript and read it myself. We are all in agreement that your study is timely and interesting and, in general, well-executed. However, I agree with the reviewers that the manuscript requires improvements. In particular, much greater clarity in the methods and results sections is required for this work to be understandable (and replicable). The reviewers also highlight a large literature on terrestrial applications of drones that has been missed in this submission. The feedback from both reviewers is very constructive and I anticipate that this manuscript will be greatly improved by carefully incorporating their comments in a resubmission.

Thank you for submitting your manuscript to RSOS.

Comments to Author:

Reviewers' Comments to Author:

Reviewer: 1

Comments to the Author(s)

Review also attached as pdf

This paper presents the results of several experiments to determine the reaction of guanaco to drones, and the variation of manual counts of these animals based on photos taken from the drone. Both of these tests were performed for a variety of drone heights and speeds of approach to the animals. The experimental design with regards to the drone operations themselves is ok, but the analysis needs more work to be of publishable quality. The methods and results section are somewhat unclear and lack necessary detail. It's true that there is not a lot of previous work in the reactions of animals to drones; and this reviewer agrees that it is likely species specific, and subject to many factors, and that we should be making note of animal reactions and adjusting observing strategies accordingly. Overall this paper does provide some new insight which may be of use to the community, provided some revisions are made.

The authors have missed a significant amount of literature on this subject. The statement in the abstract that progress so far has "mainly been derived from aquatic fauna studies" is perhaps not accurate given the wealth of other studies on terrestrial animals, such as those listed at the end of my review.

Throughout the paper the authors make incorrect use of the word "altitude". Altitude has a specific meaning, which is the height above sea level. To say you flew the drone at 60m altitude at the co-ordinates specified would likely mean you were flying underground. All instances of the word altitude should be replaced with "height" or "height AGL". I know this might not sound very scientific but it is the correct term.

Clarity of method description needs to be improved throughout. More detail is required in table 1, each flight type needs split into heights and number of photos gathered. If no photos were taken on a flight - ie it was for behaviour observation only, this should also be indicated. It is not clear if behaviour observing flights were performed before, after or at the same time as photo gathering flights. It may be that the animals became more habituated to the drone in later flights so this needs to be made clearer. The specific speeds need to be clarified for the scanning flights rather than just the range. What was the overlap in photos taken on the horizontal flights?

For the behaviour observations, how were the observations made and by how many people?

Its not clear if one of the aims of the study was to count the total number of guanacos, or if you were only interested in the number per photo. If the total number was a value of interest this should also be indicated along with details of how you account for animals moving between photos in scanning flights.

The authors mention "accuracy" in the discussion. Since no "true" ground counts were made, or at least weren't reported if they were made, it is hard to make strong statements about accuracy. The variance between different observer's counts is what is reported, and this should be made clear. Its true that variation will be related to accuracy if taking averages, but since the authors don't know if there may be a systematic offset between true and observed counts, strong statements about accuracy should be removed.

Were there any false positives identified? Given the variation in counts this seems likely. This and possible causes of false positives should be discussed.

Why was the response of the animals not monitored during VF flights? Some previous papers find that animals react strongly to a descending drone.

Was there a difference in behaviours between the wild and captive groups?

The first paragraph of the results section needs clarifying. It could read as though the count variation you are reporting is between images of the same group at different heights (I'm fairly sure that's not what you mean). Needs to be clear that the variation is between different observers counting from the same images.

Line 47 – Drones are a type of remote sensing platform, presumably you are referring to satellites here?

Line 54 – its not clear what you mean by “sampling coverage capacity” or “low flight autonomy”. These terms need to be defined more clearly or replaced.

Lines 58-61 – I'm not convinced this is true as there are many papers on terrestrial animals as well.

Line 64 – a rainforest environment is very noisy, are you sure that marine environments are louder? Are there any measurements to support this statement?

Line 111 – “y” -> ‘and’

Line 163 – Its not clear how 87% arises from 2 groups of about 50% each. Needs to be clearer where this number comes from.

Line 169 – Does this mean 14% more groups or 14% more animals?

Line 200 – similar to above, the variation has increased but you can't really make strong statements about the probability of miscounting in an absolute sense.

Line 220 – ‘motors’ not ‘engines’

Figures – more examples from different heights would help illustrate why and how it is harder to count animals at higher height AGL.

References to add

Wich, S.A.; Koh, L.P. Conservation Drones. Mapping and Monitoring Biodiversity; Oxford University Press: Oxford, UK, 2018.

Longmore, S.N.; Collins, R.P.; Pfeifer, S.; Fox, S.E.; Mulero-Pázmány, M.; Bezombes, F.; Goodwin, A.; De Juan Ovelar, M.; Knapen, J.H.; Wich, S.A. Adapting astronomical source detection software to help detect animals in thermal images obtained by unmanned aerial systems. *Int. J. Remote Sens.* 2017, 38, 2623–2638.

Mulero-Pázmány, M.; Stolper, R.; van Essen, L.D.; Negro, J.J.; Sassen, T. Remotely piloted aircraft systems as a rhinoceros anti-poaching tool in Africa. *PLoS One* 2014, 9, e83873.

- Seymour, A.C.; Dale, J.; Hammill, M.; Halpin, P.N.; Johnston, D.W. Automated detection and enumeration of marine wildlife using unmanned aircraft systems (UAS) and thermal imagery. *Sci. Rep.* 2017, 7, 45127.
- Kays, R.; Sheppard, J.; Mclean, K.; Welch, C.; Paunescu, C.; Wang, V.; Kravit, G.; Crofoot, M. Hot monkey, cold reality: Surveying rainforest canopy mammals using drone-mounted thermal infrared sensors. *Int. J. Remote Sens.* 2019, 40, 407–419.
- Van Andel, A.C.; Wich, S.A.; Boesch, C.; Koh, L.P.; Robbins, M.M.; Kelly, J.; Kuehl, H.S. Locating chimpanzee nests and identifying fruiting trees with an unmanned aerial vehicle. *Am. J. Primatol.* 2015, 77, 1122–1134.
- Bonnin, N.; Van Andel, A.; Kerby, J.; Piel, A.; Pintea, L.; Wich, S.; Bonnin, N.; Van Andel, A.C.; Kerby, J.T.; Piel, A.K.; et al. Assessment of chimpanzee nest detectability in drone-acquired images. *Drones* 2018, 2, 17.
- Wich, S.; Dellatore, D.; Houghton, M.; Ardi, R.; Koh, L.P. A preliminary assessment of using conservation drones for Sumatran orang-utan (*Pongo abelii*) distribution and density. *J. Unmanned Veh. Syst.* 2016, 4, 45–52.
- Burke, C.; Rashman, M.; Wich, S.; Symons, A.; Theron, C.; Longmore, S. Optimising observing strategies for monitoring animals using drone-mounted thermal infrared cameras. *Int. J. Remote Sens.* 2019, 40, 439–467.
- Burke, C.; Rashman, M.F.; Longmore, S.N.; McAree, O.; Glover-Kapfer, P.; Ancrenaz, M.; Wich, S.A. Successful observation of orangutans in the wild with thermal-equipped drones. *J. Unmanned Veh. Syst.* 2019, 00: 1–23 (0000) [dx.doi.org/10.1139/juvs-2018-0035](https://doi.org/10.1139/juvs-2018-0035).
- Witczuk, J.; Pagacz, S.; Zmarz, A.; Cypel, M. Exploring the feasibility of unmanned aerial vehicles and thermal imaging for ungulate surveys in forests – Preliminary results. *Int. J. Remote Sens.* 2018, 39, 5504–5521.
- Denise Spaan, Claire Burke, Owen McAree, Filippo Aureli, Coral E. Rangel-Rivera, Anja Hutschenreiter, Steve N. Longmore, Paul R. McWhirter, and Serge A. Wich. Thermal infrared imaging from drones offers a major advance for spider monkey surveys. *Drones*, 3(2), 2019.
- Chrétien, L.-P., Théau, J., and Ménard, P. 2015. Wildlife multispecies remote sensing using visible and thermal infrared imagery acquired from an unmanned aerial vehicle (UAV). *Int. Arch. Photogramm. Remote Sens. Spatial Inf. Sci.* XL-1/W4: 241–248. doi: 10.5194/isprsarchives-XL-1-W4-241-2015.
- Duffy, J. P., A. M. Cunliffe, L. DeBell, C. Sandbrook, S. A. Wich, J. D. Shutler, I. H. Myers-Smith, M. R. Varela, and K. Anderson. 2017. "Location, Location, Location: Considerations When Using Lightweight Drones in Challenging Environments." *Remote Sensing in Ecology and Conservation*. doi:10.1002/rse2.58.
- Gooday, O. J., N. Key, S. Goldstien, and P. Zawar-Reza. 2018. "An Assessment of Thermal-Image Acquisition with an Unmanned Aerial Vehicle (UAV) for Direct Counts of Coastal Marine Mammals Ashore." *Journal of Unmanned Vehicle Systems* 6 (2): 100–108. doi:10.1139/juvs-2016-0029.
- Hodgson, J. C., S. M. Baylis, R. Mott, A. Herrod, and R. H. Clarke. 2016. "Precision Wildlife Monitoring Using Unmanned Aerial Vehicles." *Scientific Reports* 6: 22574 EP. doi:10.1038/srep22574.

Israel, I., and A. Reinhard. 2017. Detecting Nests of Lapwing Birds with the Aid of a Small Unmanned Aerial Vehicle with Thermal Camera. ICUAS 2017- International Conference on Unmanned Aircraft Systems (ICUAS), Miami, FL, June 13-16.

Israel, M. 2011. "A Uav-Based ROE Deer Fawn Detection System." ISPRS - International Archives of the Photogrammetry, Remote Sensing and Spatial Information Sciences 3822: 51-55.

Evangeline Corcoran, Simon Denman, Jon Hanger, Bree Wilson & Grant Hamilton, "Automated detection of koalas using low-level aerial surveillance and machine learning", Scientific Reports volume 9, Article number: 3208 (2019)

Reviewer: 2

Comments to the Author(s)

This study expands on our understanding of how to best implement, and the disturbance caused by, the use of UAS to census and study terrestrial mammal species. Studies such as these that specifically measure and test the disturbance to animals through different flight patterns, noise levels, and altitudes are important given the burgeoning use of UAS in wildlife research. The study is well conducted and the manuscript well written. There are a few minor changes that could be made to clarify the English and I've listed these more specifically below, but overall the introduction could contextualize the proposed research in more detail compared to what has been done to date and provide more detail and clarification in the methods to improve the manuscript.

Introduction

As mentioned, most research today has been done on marine mammals and birds, however, I think more could be included here about previous research into disturbance of UAS on different species, particularly that terrestrial mammals. For example Bennitt et al. 2019, Terrestrial mammalian wildlife responses to Unmanned Aerial Systems approaches, who found varying degrees of response several terrestrial mammal species to UAS. Can you highlight the necessary advance brought by this study in contrast to previous work? Also, research such as this that are mentioned in conclusion could be raised earlier in introduction describing that studies to date suggest terrestrial mammals may respond less (Mulero-Pázmány M, Jenni-Eiermann S, Strebel N, Sattler T, Negro JJ, Tablado Z (2017) Unmanned aircraft systems as a new source of disturbance for wildlife: A systematic review. PLoS ONE 12(6): e0178448).

L48 in comparison to what? i.e. UAS are relatively less noisy than manned aircraft but not remote sensing

L51 I'm not sure I understand what is meant here, can you clarify? Perhaps that recording images using UAS offers an advantage over traditional ground surveys because the images are permanent and can be re-visited in the future for other, unforeseen analyses?

L58-59 Needs clarification and fixing for english

L61 "is" instead of "are"

Methods

L82 could you describe a bit more the enclosure particularly with relevance to their available space to react to the drones

L94 Why only 2 pictures taken at Payunia? Or is this number of pictures at each elevation for VF flights?

L104 how was group size counted? Here, and below in describing S flights, can you clarify how group size and behaviour was recorded? i.e. were there observers on foot, were the guanacos habituated to humans on the ground, or was this data collected using the imagery? If the latter,

can you provide more detail as it sounds from L105 and 106 that images were only captured if the animals did not react.

L114 can you also provide more detail on the behavioural data? i.e. data categories (forage, vigilance etc., is mentioned at the group level – was this scored based on the majority of the individuals in the group or quantified in some way i.e. number of individuals doing each behaviour on each behavioural scan?

L134-139 as above, I think this needs some clarification on the unit of analysis (i.e. one measure of behaviour per group, or individuals?)

L140 can you clarify if this is using imagery?

L150 “with VC counts at 200 m found to be more variable compared to”

Discussion

Is it possible to highlight the importance and contribution of this study in contrast to Bennitt et al. 2019’s recent analysis? Perhaps by paying more attention to inter-species differences and why reproducing these studies across a range of species and ecotypes is necessary. Given that responses may vary by species, could you also provide some discussion of the importance of these findings that may be specific to Guanacos or to disturbance potentially caused by the increasing recreational use of drones (if this is also occurring across S. America)?

L202 perhaps “aid” would make more sense here than “collaborate”?

Author's Response to Decision Letter for (RSOS-191482.R0)

See Appendix A.

RSOS-191482.R1 (Revision)

Review form: Reviewer 1

Is the manuscript scientifically sound in its present form?

Yes

Are the interpretations and conclusions justified by the results?

Yes

Is the language acceptable?

No

Do you have any ethical concerns with this paper?

No

Have you any concerns about statistical analyses in this paper?

No

Recommendation?

Accept with minor revision (please list in comments)

Comments to the Author(s)

The paper is much improved and generally scientifically sound for publication. There are only a few minor points I would like to see addressed.

Lines 53-54: "outweighed by several limitations" This is a pretty strong statement, and in fact the advantages of drones over foot surveys are already quite distinct even if some work does still need to be done. Please soften this.

Line 54: "low sampling area coverage" Relative to a foot survey even a 20-30 min drone survey can cover a much larger area, so this statement doesn't seem right.

Line 56: "time consuming data processing" Its true that processing lots of drone data by eye or by hand can be very time consuming, however is this time spent more than would be spent on a very long ground survey to make the same animal counts? The statement could perhaps be quantified better.

Line 66: "approximation" is the wrong word to use here. "Proximity" maybe?

Line 299: "results question previous mostly indirect and observational evidence" This is too strong a statement as you have only examined one species of animal. Its an important new observation but in context with the rest of the literature I think more evidence is needed to make a statement that strong.

Review form: Reviewer 2

Is the manuscript scientifically sound in its present form?

Yes

Are the interpretations and conclusions justified by the results?

Yes

Is the language acceptable?

Yes

Do you have any ethical concerns with this paper?

No

Have you any concerns about statistical analyses in this paper?

No

Recommendation?

Accept with minor revision (please list in comments)

Comments to the Author(s)

The changes made to the manuscript have better contextualized this research with what has been done to date and improved the clarity of the objectives and methods used. I have included just a few more comments below mostly regarding clarity and English.

L23 this sounds as though the studies themselves are a concern – perhaps better to just say “Few studies to date have experimentally evaluated the effect of UAS approach on animals and have usually focused primarily on aquatic fauna”

L29 “limit disturbance of animals and flying low enough to maintain count precision...”

L58 “Moreover, images provide a permanent recording of data that can be re-visited and analysed again in the future, unforeseen research questions.”

L67 different types

L70 “However, few studies have experimentally... animals, whilst those studies done to date have focused primarily...”

L78 Include the authors names here and put the citation at the end of the relevant text., same for 22 and 23

L99 whilst maintaining count precision

L134 english needs amending, perhaps “We approached the drone at 200 AGL until it was positioned directly above a previously selected group of guanacos. Then the drone was slowly lowered and pictures were taken at...”

L139 approached the drone towards it

L149 To be absolutely clear state whether group size was recorded by the observer here.

L189-194 Were these behaviours recorded as counts of individuals doing each behaviour, as the predominant behaviour displayed by most of the group, or by focal individuals by the observer?

L210 were not was – perhaps clearer to say “At 100 and 150m AGL only a small number of photos were captured that contained offspring, so we could not include these images in the analysis.

L229 14% more groups reacted when the UAS was flown at 8-10m/s.

L277 has been shown to

L287 “which supports previous research suggesting other terrestrial mammals (which ones?) also react more to auditory disturbance”

L298 spelling usually

L318 include author names and move citation to end of text if structuring sentences this way

L324 highlights

L328 especially not specially

L329-331 This sentence is very unclear, can you check and rephrase?

Table 1. why is flight height presented 60/180 for Feb 2018?

Figure 3. still says altitude instead of height

Decision letter (RSOS-191482.R1)

27-Nov-2019

Dear Dr Schroeder,

On behalf of the Editors, I am pleased to inform you that your Manuscript RSOS-191482.R1 entitled "An experimental approach to evaluate the potential of drones in terrestrial mammal research: a gregarious ungulate as a study model" has been accepted for publication in Royal Society Open Science subject to minor revision in accordance with the referee suggestions. Please find the referees' comments at the end of this email.

The reviewers and Subject Editor have recommended publication, but also suggest some minor revisions to your manuscript. Therefore, I invite you to respond to the comments and revise your manuscript.

- Ethics statement

- Data accessibility

If you wish to submit your supporting data or code to Dryad (<http://datadryad.org/>), or modify your current submission to dryad, please use the following link:
<http://datadryad.org/submit?journalID=RSOS&manu=RSOS-191482.R1>

- Competing interests

- Authors' contributions

- Acknowledgements

- Funding statement

Please note that we cannot publish your manuscript without these end statements included. We have included a screenshot example of the end statements for reference. If you feel that a given

heading is not relevant to your paper, please nevertheless include the heading and explicitly state that it is not relevant to your work.

Because the schedule for publication is very tight, it is a condition of publication that you submit the revised version of your manuscript before 06-Dec-2019. Please note that the revision deadline will expire at 00.00am on this date. If you do not think you will be able to meet this date please let me know immediately.

Kind regards,

Lianne Parkhouse
Editorial Coordinator
Royal Society Open Science

on behalf of Dr Alecia Carter (Associate Editor) and Professor Kevin Padian (Subject Editor)
openscience@royalsociety.org

Reviewer comments to Author:

Reviewer: 1

Comments to the Author(s)

The paper is much improved and generally scientifically sound for publication. There are only a few minor points I would like to see addressed.

Lines 53-54: "outweighed by several limitations" This is a pretty strong statement, and in fact the advantages of drones over foot surveys are already quite distinct even if some work does still need to be done. Please soften this.

Line 54: "low sampling area coverage" Relative to a foot survey even a 20-30 min drone survey can cover a much larger area, so this statement doesn't seem right.

Line 56: "time consuming data processing" Its true that processing lots of drone data by eye or by hand can be very time consuming, however is this time spent more than would be spent on a very long ground survey to make the same animal counts? The statement could perhaps be quantified better.

Line 66: "approximation" is the wrong word to use here. "Proximity" maybe?

Line 299: "results question previous mostly indirect and observational evidence" This is too strong a statement as you have only examined one species of animal. Its an important new observation but in context with the rest of the literature I think more evidence is needed to make a statement that strong.

Reviewer: 2

Comments to the Author(s)

The changes made to the manuscript have better contextualized this research with what has been done to date and improved the clarity of the objectives and methods used. I have included just a few more comments below mostly regarding clarity and English.

L23 this sounds as though the studies themselves are a concern – perhaps better to just say “Few studies to date have experimentally evaluated the effect of UAS approach on animals and have usually focused primarily on aquatic fauna”

L29 “limit disturbance of animals and flying low enough to maintain count precision...”

L58 “Moreover, images provide a permanent recording of data that can be re-visited and analysed again in the future, unforeseen research questions.”

L67 different types

L70 “However, few studies have experimentally... animals, whilst those studies done to date have focused primarily...”

L78 Include the authors names here and put the citation at the end of the relevant text., same for 22 and 23

L99 whilst maintaining count precision

L134 english needs amending, perhaps "We approached the drone at 200 AGL until it was positioned directly above a previously selected group of guanacos. Then the drone was slowly lowered and pictures were taken at..."

L139 approached the drone towards it

L149 To be absolutely clear state whether group size was recorded by the observer here.

L189-194 Were these behaviours recorded as counts of individuals doing each behaviour, as the predominant behaviour displayed by most of the group, or by focal individuals by the observer?

L210 were not was - perhaps clearer to say "At 100 and 150m AGL only a small number of photos were captured that contained offspring, so we could not include these images in the analysis.

L229 14% more groups reacted when the UAS was flown at 8-10m/s.

L277 has been shown to

L287 "which supports previous research suggesting other terrestrial mammals (which ones?) also react more to auditory disturbance"

L298 spelling usually

L318 include author names and move citation to end of text if structuring sentences this way

L324 highlights

L328 especially not specially

L329-331 This sentence is very unclear, can you check and rephrase?

Table 1. why is flight height presented 60/180 for Feb 2018?

Figure 3. still says altitude instead of height

Author's Response to Decision Letter for (RSOS-191482.R1)

See Appendix B.

Decision letter (RSOS-191482.R2)

06-Dec-2019

Dear Dr Schroeder,

It is a pleasure to accept your manuscript entitled "An experimental approach to evaluate the potential of drones in terrestrial mammal research: a gregarious ungulate as a study model" in its current form for publication in Royal Society Open Science.

Best regards,

on behalf of Dr Alecia Carter (Associate Editor) and Kevin Padian (Subject Editor)
openscience@royalsociety.org

Appendix A

8-Oct-19

Response to Referees

First of all, the authors would like to thank the Associate Editor and referees for their comments, which have contributed to a notable improvement of our manuscript. Below, we respond in detail to each of the suggestions and comments.

Associate Editor's comments (Dr Alecia Carter):

We have included more detail in methods and results following each suggestion made, to give greater clarity and replicability to the manuscript. We have substantially changed the introduction, and also parts of the discussion, to incorporate information from the literature on UAS applications in terrestrial fauna suggested by the reviewers.

Reviewers' Comments to Author:

Reviewer: 1

General comments

1. Thanks to the reviewer for the list of literature provided. We have substantially changed the introduction, and also parts of the discussion, to incorporate information from the literature on UAS applications in terrestrial fauna suggested by the reviewer. We have therefore focused on contrasting the progress achieved in evaluating the technical feasibility of the use of drones in wildlife in general, with the little progress in understanding the reaction of animals, highlighting mainly the few experimental studies that exist to date on terrestrial mammals. We have also modified the abstract according to those changes.
2. We have replaced the word "altitude" with the word "height" and "height AGL" throughout the text.

3. As suggested, for Table 1 we divided the flights by height and photos taken, clarifying the flights in which no photos were recorded because they were for behaviour only. In the text we include a clarification about the precautions we had at the time of doing the experiments seeking not to fly over the same group more than once, and thus avoid the habituation of the animals. Also, we included the specific speeds of the scanning flights. In the horizontal flights (HF) the photos were taken manually on the groups that did not react. No overlap is established in that type of flights. The overlap in photos was programmed in the S and Sb flights and, as it is expressed in the text, was 20%, the minimum possible, since the photos were used individually, not as part of an orthomosaic.

4. We included a detailed description of how the observations were made and by how many people.

5. The aim of the study was not to count the total number of guanacos; we were only interested in the number per photo. This was so because our goal was to know if the ability to record adults and offspring in the photos varied with flight height. See also the comment 6 below. The scanning flights were made to obtain individual photos that would then be used in objective 1 for counting (S, Sb), or recording behaviour (Sb).

6. Following the reviewer's observation, with which we agreed, we have removed or changed the word "accuracy" from the text when we refer to objective 1 and its derived results or discussions.

8. We agreed with the reviewer that given the variability in counts, there were surely false positives, as well as false negatives. But given that the observers gave us a spreadsheet with the counts they observed per photo, and not the photos with the individuals marked on them, we cannot identify exactly in which photos there were. Also, as the reviewer points out, we didn't do "real" counts in the field to get a real count of animals, because that wasn't our goal. We did not make comparisons of counts referring to a real value, but we were interested in analysing the variability of the

counts among themselves. However, the reviewer's comment seems relevant to us and we include some lines of discussion on this point in the second paragraph of the discussion section.

9. The reviewer is right. There is literature that found a strong reaction to the descending flight of the drone. In our case, as the VF flights were not used for the behavioural objective, but for the counting one, we did not record behaviour during those flights. We could not descend less than 150 m during flights performed on wild animals, since they reacted to the drone and escaped. Instead, the captive population allowed us to descend to 50 m (Table 1). Probably, there must be a kind of habituation on the captive population to disturbances, but that was not the objective of our work; we used the captive population to guarantee having photos of the animals taken at low heights. This relates to the reviewer's next question: "*Was there a difference in behaviours between the wild and captive groups?*". It was not an objective of this work to evaluate the difference in behaviour between the captive and the wild population, something that would be interesting to evaluate in future studies. Behavioural experiments were only performed in the wild population (figure 1, table 1).

10. In the first paragraph of the results section we clarify that the count variation is between different observers counting from the same images.

Line 47 - Although we were referring to satellite images, we removed this reference as it does not qualify as a "classic methods used in wildlife research", but satellite images are counting methods that are being tested recently.

Line 54 - As suggested, we included clarifications referred to the low sampling area coverage and autonomy of the small drones.

Lines 58-61 - We completely change this paragraph of the introduction, interpreting the literature on terrestrial mammals suggested by the reviewer.

Line 64 - We modified this part of the introduction.

Line 111 - We changed “y” for ‘and’

Line 163 - We clarified that 87% of the groups reacted, but of those that reacted (100%), 51.2% reacted at 2-4 m/s, while 48.8% reacted at 8-10 m/s.

Line 169 - We clarified there was a 14% more reaction of groups (not animals) at rapid speed.

Line 200 - We changed the original sentence to consider the reviewer’s comments

Line 220 - We changed ‘engines’ for ‘motors’

Figures - We included more examples of photos from different heights to help illustrate why and how it is harder to count animals at higher height
AGL

Reviewer: 2

Comments to the Author(s)

Introduction

According to the reviewer’s comments, we have changed the introduction to contextualize our research in more detail compared to what has been done to date. We included more information from previous research into disturbance of UAS on different species, particularly for terrestrial mammals. Specifically, we highlight the necessary progress that our study represents in contrast to the previous ones, such as Mulero-Pazmay et al 2017, Bennitt et al. 2019 and others.

L48. We removed remote sensing as it does not qualify as a "classic methods used in wildlife research", and included "UAS are safe for operators, relatively less noisy than manned aircraft..." .

L51. Yes, we mean that recording images using UAS offers an advantage over traditional ground surveys because the images are permanent and can be re-visited in the future for other, unforeseen analyses. We clarified the sentence, including at the end: "... as images can be checked and analysed several times"

L58-59. We removed this sentence of the introduction, and we completely changed the paragraph.

L61 We completely changed the paragraph.

Methods

L82 . We describe the enclosures of the Los Peucos farm to show the available space that guanacos have to react to drones.

L9. We took only two photos in La Payunia (one at 150 m and another at 200 m) in VF flights, mainly because the groups in this wild population are quite reactive (as we later verified with our behaviour analysis) and it is difficult to reach and descending on them without getting them scared. But since the VF flights were performed for the counting objective, and not for the behaviour objective, we completed the amount of photos in Payunia with the S and Sb flights. It seems important to emphasize that for the purpose of counting, we seek to have as many photos as possible of adults and offspring, at different heights, beyond the population to which they belong.

L104. We included a detailed description of how group size and behaviour was recorded, and of the precautions we had at the time of doing the experiments to avoid the habituation of the animals.

L114. We clarified that behavioural data categories (forage, vigilance, locomotion, others) were scored for each of the members of the group using a scan sampling methodology.

L134-139. We included the sentence “calculated as the number of animals displaying each behavioural category divided the total number of animals within each group” to clarify the unit of analysis for Sb behavioural data.

L140. We clarified that neither group size nor behaviour was taken from the images, but by means of an observer (new lines 124-136).

L150. As suggested, we included the phrase: “with VC counts at 200 m found to be more variable compared to”

Discussion

Following the suggestion, in the last paragraph of the discussion we highlighted the contribution of our study in contrast with the recent analysis of Bennitt et al. 2019 and we pointed out the importance of our findings for guanaco, associated to potential disturbance that could bring the recreational use of UAS, still little developed in South America, but in expansion worldwide.

L202. We changed the word “collaborate” for the word “aid”.

Appendix B

Nov-9-19

Response to Referees

Again, we would like to thank the Associate Editor and referees for their comments, which are answered in detail below. In addition, we had the entire manuscript reviewed by an English-speaking colleague. Following this colleague's suggestion, we changed "Variation coefficient (VC)" to "Coefficient of Variation (CV)" throughout the text, as it is the most commonly used term in literature.

Reviewers' Comments to Author:

Reviewer: 1

Lines 53-54: Regarding this and the following two comments, we changed the paragraph substantially, to soften this statement.

Line 54: We removed this sentence

Line 56: We removed this sentence

Line 66: We changed "approximation" for "proximity."

Line 299: We changed "results question previous mostly indirect and observational evidence" for "results *call into* question previous mostly..." in order to soften the sentence.

Reviewer: 2

L23 We changed the sentence according to the reviewer's suggestion

L29 We changed the sentence according to the reviewer's suggestion

L58 We changed the sentence according to the reviewer's suggestion

L67 We changed for “different types”

L70 We changed the sentence according to the reviewer’s suggestion

L78 For citations 21, 22 and 23 the names of the authors were included and the citation was placed at the end of the relevant text.

L99 We changed the sentence according to the reviewer’s suggestion

L134 We changed the sentence according to the reviewer’s suggestion

L139 We used “flew the drone towards it”

L149 We stated that group size was recorded by the observer.

L189–194 We clarified this point, stating that “We observed all the animals as a focal group [29]” (lines 119–120) and that “We considered a flight reaction when at least one of the individuals of the focal group behaved that way, followed by the others (lines 174–175).

L210 We changed the sentence according to the reviewer’s suggestion.

L229 We changed the sentence according to the reviewer’s suggestion

L277 We used “has been shown to”

L287 We changed the sentence according to the reviewer’s suggestion.

L298 We used “usually”

L318 We included author names and moved citation to the end of text.

L324 We maintain “highlight” because we are referring to “these results”

L328 We used “especially”

L329–331 We rephrased the sentence.

Table 1. We changed the presentation of the heights of HF for Feb 2018, but we also standardized the records in S and Sb, so as not to repeat heights. We believe that this way the table is better explained.

Figure 3. We used “height”